# Cobalt Incorporated Graphitic Carbon Nitride as a Bifunctional Catalyst for Electrochemical Water-Splitting Reactions in Acidic Media

**DOI:** 10.3390/molecules27196445

**Published:** 2022-09-30

**Authors:** Shibiru Yadeta Ejeta, Toyoko Imae

**Affiliations:** 1Graduate Institute of Applied Science and Technology, National Taiwan University of Science and Technology, Keelung Road, Taipei 10607, Taiwan; 2Department of Chemical Engineering, National Taiwan University of Science and Technology, Keelung Road, Taipei 10607, Taiwan

**Keywords:** bifunctional catalyst, graphitic carbon nitride, metallic cobalt, electrochemical water-splitting, hydrogen evolution reaction, oxygen evolution reaction

## Abstract

Non-noble metal-based bifunctional electrocatalysts may be a promising new resource for electrocatalytic water-splitting devices. In this work, transition metal (cobalt)-incorporated graphitic carbon nitride was synthesized and fabricated in electrodes for use as bifunctional catalysts. The optimum catalytic activity of this bifunctional material for the hydrogen evolution reaction (HER), which benefitted at a cobalt content of 10.6 wt%, was promoted by the highest surface area and conductivity. The activity achieved a minimum overpotential of ~85 mV at 10 mA/cm^2^ and a Tafel slope of 44.2 mV/dec in an acidic electrolyte. These values of the HER were close to those of a benchmark catalyst (platinum on carbon paper electrode). Moreover, the kinetics evaluation at the optimum catalyst ensured the catalyst flows (Volmer–Heyrovsky mechanism), indicating that the adsorption step is rate-determining for the HER. The activity for the oxygen evolution reaction (OER) indicated an overpotential of ~530 mV at 10 mAcm^−2^ and a Tafel slope of 193.3 mV/dec, which were slightly less or nearly the same as those of the benchmark catalyst. Stability tests using long-term potential cycles confirmed the high durability of the catalyst for both HER and OER. Moreover, the optimal bifunctional catalyst achieved a current density of 10 mAcm^−2^ at a cell voltage of 1.84 V, which was slightly less than that of the benchmark catalyst (1.98 V). Thus, this research reveals that the present bifunctional, non-noble metallic electrocatalyst is adequate for use as a water-splitting technology in acidic media.

## 1. Introduction

Energy is remarkably vital to the development of the global economy. However, the overuse of fossil fuel energy is objectionable for the environment. Thus, the development of sustainable, green, and environmentally friendly alternative energy sources is essential, and humans are constantly seeking such sources [1]. One of these alternative green energy sources is the hydrogen energy [2,3,4]. A water-splitting generator received great attention for producing molecular hydrogen because hydrogen reveals high energy density (143 KJ/g), plentiful availability, and zero carbon emission [5,6]. Although hydrogen can be produced from hydrogen-containing compounds (biomass, water, natural gas, etc.), hydrogen from water electrolysis is the cleanest and most sustainable method [7,8]. In such cases, hydrogen evolution reaction (HER) and oxygen evolution reaction (OER) occur at the cathode and anode half-reactions, respectively, and the method opens new strategies for obtaining electric energy from the most abundant resource (water) [9]. However, both HER and OER must overcome a large energy barrier (overpotential). Thus, the scalable industrial application is still under experiment, and until now, only a small number (about 4%) of companies can produce hydrogen by water electrolysis [8]. OER in particular requires a large overpotential due to a four-step oxidation process. To overcome these barriers, researchers are trying to find efficient electrocatalysts that require low overpotential for both half reactions and are sufficiently stable for long-term use [10].

Mainly Pt, Pd and RuO_2,_ or IrO_2_ are used as active benchmark electrocatalysts for HER and OER, respectively, but their scarcity, poor stability, and high cost hinder their use in large-scale applications [10]. Thus, developing cost-efficient, highly active, and stable alternative catalysts from more abundant raw materials is required for large-scale electrochemical water-splitting devices. Another possible approach to minimize cost is the development of highly efficient bi-functional electrocatalysts with low overpotential to drive the catalytic reaction, owing to their integrated merits for simplifying device fabrication and reducing cost [11]. The feasibility of bi-functional catalysts highly depends on the performance of the electrocatalysts to minimize overpotential in such reactions. To date, most bifunctional catalysts were reported in alkaline mediums, and very few were reported in acidic mediums [12]. The pursuit of reactions at common pH conditions for both HER and OER is also an issue for simplifying the device. Different catalysts with diverse merits have been reported for HER [12,13,14]. The minimization of the quantity of precious metals can be realized using a carbon-based material holder with excellent stability and enhanced conductivity [1,15]. These carbon-based materials can decrease the energy adsorption of hydrogen and increase its conductivity, enhancing the activity of HER. In addition, replacing those metals with abundant transition metals on graphene or graphitic carbon has been also reported [16]. Transition metal-based particles, such as Co, Ni, Fe, and Cu, or their alloys, oxides, phosphides, and carbides encapsulated in carbon materials have been proposed [11,17]. Different functional (bimetallic, trimetallic, or high-cost) materials were also reported for HER and OER, but in most cases, such reactions were performed differently in alkaline and acidic pH conditions [16,18,19,20].

The physicochemical properties of graphitic carbon nitride (g-CN) make it a striking component for constructing an alternative catalyst [17]. Although the pristine g-CN has poor conductivity, this shortcoming can be overcome by integrating electron conductive materials onto it. The encapsulation of metals is due to the pyridinic nitrogen in g-CN that can attract the metal ions/atoms and coordinate them in the porous structure [16]. In this work, the composites, consisting of cobalt encapsulated in g-CN, were synthesized by chemical reduction and used as bifunctional catalysts for the HER activity. Furthermore, the optimized electrode was also employed for electrocatalytic OER by changing the potential only in the same electrolyte. The quality of the present optimized catalyst was compared with a benchmark catalyst and previously reported bifunctional catalysts regarding the efficiency of hydrogen fuel cells.

## 2. Materials and Methods

### 2.1. Reagents and Compounds

Melamine powder (99%), sodium borohydride (NaBH_4_, 98%), and cobalt(II) chloride anhydrous (CoCl_2_) were obtained from Across Organics (New Jersey, NY, USA). Carbon paper substrate (0.44 g cm^−3^, 0.19 mm thickness) was purchased from Toray Company (Tokyo, Japan). Deionized water (resistivity: 18.2 MΩ·cm) was used for the preparation of all solutions. Other chemicals were of commercial reagent grades and were used without further purification.

The g-CN powder is the same sample that was synthesized and characterized in a previous work [21]. It was synthesized by pyrolysis of melamine and exfoliation by strong sonication. A composite of cobalt on g-CN (Co@g-CN) was synthesized by the reduction of the cobalt precursor in a solution with sodium borohydride (NaBH_4_): g-CN (50 mg) was dispersed in water (1.6 g L^−1^, 30 mL) by sonicating for 1 h, and an aqueous CoCl_2_ solution (0.01 M, 2.5, 5, 10, or 15 mL) was added to the g-CN dispersion and stirred overnight. Subsequently, an aqueous NaBH_4_ solution (0.5 M, 2 mL) was added and further stirred for 1 h. The solid product was collected by filtering with a filter paper (2 µm pore). The calculated contents of cobalt in Co@g-CN were 2.9, 5.6, 10.6, and 15.0 wt%, respectively, depending on the added volume of the CoCl_2_ solution. The products were then termed as Co(x)@g-CN (x = 3, 6, 11, 15).

### 2.2. Instruments

A photoluminescence (PL) spectroscope (F-7000, Hitachi High-Technologies Co., Tokyo, Japan), an ultraviolet (UV)-visible absorption spectrometer (JASCO V-670, Tokyo, Japan), and a Fourier transform infrared (FTIR) absorption spectrometer (NICOLET 6700, Thermo Scientific, Waltham, MA, USA) were used for spectrometric measurements using the diffuse reflectance Fourier transform mode. An X-ray diffraction was recorded on a diffractometer (Bruker, D2 phaser, Bremen, Germany) using CuKα radiation (1.54 Å) at 40 kV and 30 mA, a transmission electron microscope (TEM, JEOL, Tokyo, Japan) was operated at an accelerating voltage of 120 kV, nitrogen adsorption-desorption isotherm experiments were caried out at 77 K using a Brunauer-Emmett-Teller (BET) surface area analyzer (BELSORP Max, Osaka, Japan), and X-ray photoelectron spectroscopic (XPS) measurements were performed on a VG scientific ESCALAB 250, England.

The linear scan voltammetry (LSV) measurements were performed using a computer-controlled potentiostat on a Zahner electrochemical workstation (Zennium, 40442, Kronach, Germany) with a three-electrode system: a cell consists of an aqueous 0.5 M H_2_SO_4_ solution (25 mL) as an electrolyte, a platinum wire as a counter electrode, an Ag/AgCl in 3 M KCl as a reference electrode, and a working electrode, which was prepared on a carbon paper by loading a catalyst ink (slurry, 2.0 ± 0.2 mg·cm^−2^, 50 μL), that is, an ultrasonically dispersed and stirred mixture of Co(x)@g-CN (10 mg), isopropanol (200 μL), and Nafion binder (0.5 vol%, 100 μL). A working electrode of a benchmark catalyst, called Pt/Cp, was prepared by coating Pt of 1 nm thickness by 30 min vapor deposition on a surface of carbon paper with a 1 cm^2^ active area. Electrochemical impedance spectroscope (EIS) measurement was performed using the same electrode system and electrolyte, at a potential of −0.25 V, a frequency range of 10 kHz–1 MHz, and an amplitude of 10 mV on the same instrument.

The HER and OER activities were recorded by linear sweep voltammetry (LSV) at a scan rate of 10 mV·s^−1^, and the electrode durability was tested by a cyclic voltammogram (CV) response. Potentials were converted to the reversible hydrogen electrode (RHE) scale as denoted by E_RHE_. Then CVs were conducted at a scan rate of 10 mV/s between 0.10 to −0.11 V vs. E_RHE_, where a quasi-rectangular shaped curve was obtained. The current density was calculated by $J=\frac{I}{A}$, where I is a current and A is an active area of an electrode. The overpotential (η) was calculated according to the following formula: η [V] = 0.000 − E_RHE_ for HER and η [V] = E_RHE_ − 1.230 for OER. Then, the slope b of the Tafel plot was calculated from a Tafel equation (Equation (1)).
η = a + b log |J|(1)
where a is an overpotential at current density (J = 1).

The hydrogen and oxygen gases generated at respective electrodes were quantified, using a laboratory-made instrument, by the downward displacement of water in which electrodes prepared as the cathode (HER), anode (OER), and Pt as a reference electrode were used at an applied potential of 1.8 V.

## 3. Results and Discussion

### 3.1. Structural Characteristics of Co@g-CN

On a 2D g-CN sheet synthesized by a pyrolysis condensation reaction of melamine [21], Co was deposited via the reduction reaction of Co precursor (See Figure 1A). FTIR spectra of Co@g-CN with different Co contents were measured and compared with g-CN (Figure 1B). The characteristic IR bands of g-CN are a stretching N-H vibration mode of the terminal amine at 3100–3300 cm^−1^, C=N, C-N, N-C_3_, and NH-C_2_ stretching vibration modes of the heptazine unit at 1640, 1410, 1322, and 1250 cm^−1^, respectively, and an out-of-plane bending vibration mode of the heptazine unit at 810 cm^−1^ [21]. These bands were observed even after loading the Co, and the relative band intensities were also nearly unchanged. These results indicate that the loading of Co does not disturb the vibrational modes of g-CN with a heptazine unit ring consisting of different carbon-nitrogen bonds.

A crystal structure of Co@g-CN was analyzed using an XRD pattern. As shown in Figure 1C, Co(11)@g-CN exhibited two distinctive diffraction peaks at 2θ = 27.99° and 13.36°, in good agreement with those of g-CN. A weak peak indexed as (100) equivalent to d spacing of 0.662 nm can be ascribed to the in-plane distance of the g-CN sheet, and a strong peak (002) with d spacing of 0.318 nm is attributed to the interlayer distance between the g-CN sheets [21]. However, it must be noted that no peaks of cobalt species in Co(11)@g-CN were observed, although they were observed at 2θ = 44, 51, and 75° [22]. The reason should be the low content of Co [23,24]. However, the decrease in intensity and the slight broadening of (002) plane indicate the coordination of Co to g-CN by the host-guest framework forming Co-N bonds [10]. From these results, even though the FTIR spectra did not confirm the existence of Co, XRD spectra indicated the coordination of Co in the framework of g-CN.

On an XPS spectrum of Co(11)@g-CN, as seen in a survey spectrum of Figure 2A, additional peaks of the Co element were observed, as well as C, N, and O elements, indicating the loading of Co on a structure of g-CN. The fine spectra and their deconvoluted peaks of each element are shown in Figure 2B. Co 2p exhibited two intense binding peaks at 783.1 and 799.8 eV, which are attributed to 2p_3/2_ and 2p_1/2_ of cobalt, respectively. These binding energies are higher than those of free metallic cobalt and cobalt oxide [23,24], but they can be attributed only to metallic cobalt coordinated to the center of the nitrogen atoms in the heptazine ring (Co-N_x_) [10], as illustrated in Figure 1A. Meanwhile, the peaks at 787.7 and 805.9 eV are satellite peaks of 2p_3/2_ and 2p_1/2_ peaks, respectively. The result shows that Co is successfully incorporated in the structure of the graphitic framework. In comparison with XPS peaks of g-CN [21], the same number of deconvoluted peaks of C 1s and N 1s were observed, but both the C 1s and N 1s binding energies of the C=N-C bond and an N 1s binding energy of C-NH_x_ bond were remarkably shifted from the corresponding binding energies of g-CN [21] because of the influence of the Co-N_x_ coordination on the binding energies of NC conjugation. Additionally, O 1s was deconvoluted in three peaks in comparation to one peak of g-CN at 287.2 eV [21]. The new peaks are attributed to CO_2_ and H_2_O being simply adsorbed to Co@g-CN. CO_2_ is easily adsorbed on the amine group and forms the NHCOO^−^ group based on an equation of CO_2_ + 2 R-NH_2_ ⇌ R-NHCOO^−^ + R-NH_3_^+^ [25]. Then, peaks of the NHCOO^−^ group were found in C 1s and O 1s, but C 1s of CO_2_ was assumed to overlap on the C1s of any group. Thus, the XPS spectra confirmed that Co was incorporated in the graphitic framework through Co-N_x_ bonding [24].

A TEM image of Co(11)@g-CN in Figure 3a showed the sheet-like structure of g-CN [21] and the grain texture with an average size of 11.72 ± 2.56 nm on the sheet. An EDS on the FE-SEM image of Co(11)@g-CN (Figure 3b,c) confirmed the existence of the Co atom in Co(11)@-gCN. Moreover, an inset table in EDS indicates the existence of 11.6 wt% Co. This value is comparable to the calculated value (10.6 wt%) in Co(11)@-gCN. Moreover, the atom ratio of C:N in g-CN, which can be evaluated from Figure 1A, is 18.0:25.5. Since the atom ratio of C:N: Co calculated from the atom % in Figure 3c (inset) was 18.0:26.9:0.77; Co(11)@g-CN achieved 77% Co loading of the estimated complex structure.

As seen in Figure 4a, nitrogen adsorption-desorption curves of Co(x)@g-CN (x = 6, 11, and 15) at different cobalt contents belonged to the type II isotherm, showing hysteresis at high relative pressure due to the porous structure of the materials. The specific surface area, pore volume, and pore diameter were analyzed from the BET isotherms, and the Barret, Joyner, and Halenda (BJH) plots are shown in Figure 4b. As seen in Table 1, Co(11)@g-CN exhibited the largest specific surface area and pore volume and the smallest pore diameter compared to Co(x)@g-CN with other Co contents. These results show the preferable variation of the surface properties of g-CN with the addition of cobalt at the optimum content of 10.6 wt%.

On a UV-visible absorption spectrum of Co(11)@g-CN, in comparison with that of g-CN (Figure 5a), while the main and shoulder bands of g-CN were seen at 331 nm and around 400 nm, respectively, an absorption band at a shorter wavelength (330 nm) of Co(11)@g-CN decreased its intensity. These absorption bands can be assigned to π-π* and n-π* electronic transitions of C=N-C in the heptazine ring and the free electron of 2p [26]. Thus, the variation in the intensity after the addition of Co may be attributed to the formation of the Co-N interaction, which can decrease the intensity of the π-π* band. The influence of Co loading on the optical property of g-CN was observed on the fluorescence spectra, where an emission band at 440 nm with a shoulder band around 460 nm was observed at a 275 nm excitation for both g-CN and Co@g-CN, as shown in Figure 5b. While supplementary excitation bands were detected at 330 and 360 nm, the relevant emission bands displayed quite weak intensities at 440 nm. It should be noted that the fluorescence intensities of g-CN at 275 nm (excitation) and 440 nm (emission) reduced after the incorporation of metallic Co. Since a 275 nm band can be assigned the π-π* electron transition, the fluorescence property is related to the π-π* transition, but not the n-π* transition. This decrease in fluorescence may participate in the energy transfer [27,28]. The transferred energy contributes to the charge separation, promoting the electron mobility within the structure of the graphitic carbon nitride. Incidentally, the band gap (2.60 eV) of g-CN minimized to 2.38 eV in Co(11)@g-CN, as evaluated from the UV-visible absorption spectra (Figure 5c), indicated that metal doping decreases the bandgap of Co(11)@g-CN.

### 3.2. Electrochemical Water-Splitting Activity of Co@g-CN Catalysts

It was revealed from the characterization of Co@g-CN that Co incorporated in g-CN varies surface properties of g-CN and enables the charge transfer from g-CN. These effects should influence the water adsorption and the increase in the conductivity, which can enhance the catalytic activity of the HER and OER. Thus, the electrochemical activities of catalysts (Co@g-CN) towards the HER and OER were examined using a three-electrode system in an acidic electrolyte medium (0.5 M H_2_SO_4_ (aq.). The HER activities of the prepared catalysts and a benchmark catalyst (Pt/Cp) are presented in Figure 6a. The overpotentials of catalysts towards HER activities at a standard current density (10 mA·cm^−2^) are summarized in Table 2. Although g-CN exhibited poor electrocatalytic performance for HER, the incorporation of the small amount of Co on the g-CN sheet decreased the overpotential. The Co(11)@g-CN required an overpotential of 85 mV vs. E_RHE_ to achieve a current density of 10 mA·cm^−2^, although the overpotential of Co(15)@g-CN was the same as that of a benchmark catalyst (65 mV).

A Tafel plot obtained from LSV of HER can be employed for estimating the rate-determining step [1,15]. Figure 6b displays Tafel slope plots of g-CN, Co@g-CN, and Pt/Cp electrocatalysts, and numerical values of the slope are listed in Table 2. The experimental slope of Co(11)@g-CN was the smallest for Co(x)@g-CN and similar to that of a benchmark catalyst [29]. The theoretical Tafel slope values are ~120, ~40, and ~30 mV/decade, corresponding to Volmer, Heyrovsky, and Tafel reactions, respectively [1]. Compared to the theoretical values, the experimental slope of Co(11)@g-CN, as well as the benchmark catalyst, follows the Volmer–Heyrovsky step in the HER. That is, in the first step, H^+^ in the electrolyte and e^−^ form an adsorbed H atom (H_ads_) on an electrocatalytic active site [30], and in the second step, the H_ads_ reacts with H^+^ and e^−^ to yield H_2_ through an electrochemical desorption process [11].
H^+^ + e^−^ → H_ads_ (Volmer)(2)
H_ads_ + H^+^ + e^−^ → H_2_ (Heyrovsky)(3)

On the other hand, the OER electrocatalyst is an energy-uphill reaction involving four-electron transfer processes in an acidic environment, but the correlation among the kinetics equations is extremely complicated to solve [11]. Therefore, when Co(11)@g-CN and Pt/Cp electrocatalysts were applied for OER activity, overpotentials of 530 and 680 mV, respectively, were obtained at 10 mA cm^−2^ (see Figure 6c and Table 2). The consistency of the OER Tafel slopes between Co(11)@g-CN and Pt/Cp catalysts was also achieved, as seen in Figure 6d and Table 2.

EIS measurements were performed to further characterize Co@g-CN according to conductivity. Nyquist plots obtained from EIS of Co@g-CN are shown in Figure 6e, and resistances were evaluated using a Randles circuit model, as listed in Table 2. Co(11)@g-CN showed the lowest charge transfer resistance (Rct) of 27.0 Ω, which was small compared to g-CN (6.06 kΩ), indicating that the incorporation of Co into g-CN enhanced the conductivity, although the variation of the internal resistance (Rs) was less, depending on the content of Co. However, the excess addition of Co inhibited the conductivity increase, showing a certain limit for the advancement of activity. The stability of an electrocatalyst is another important factor for its suitability for practical applications. Thus, the LSV polarization measurement of Co(11)@g-CN was performed after 2000 continuous cycles (scan rate: 10 mV·s^−1^) for both HER and OER. The observed catalytic decay compared to the initial activity was very small (see Figure 6f,g), indicating better stability, and therefore, the effective durability for repeatedly using the Co(11)@g-CN electrocatalyst in acidic media.

The amounts of hydrogen and oxygen arising over time were measured using a setup shown in Figure 7A. The volume of hydrogen gas increased with time, and 1.5 mL was collected at 7 h; similarly, the volume of oxygen gas also increased with time in an equimolar ratio, as theoretically expected (Figure 7B), indicating the high performance of the present catalyst (Co(11)@g-CN).

Thus, the catalytic activity of g-CN was enhanced in the presence of metallic cobalt. This activity demonstrates that the incorporation of metallic Co in g-CN influences the HER. Specifically, Co coordinates chemically with N atoms in the heptazine moieties and generates Co-N active sites, which significantly increases HER and OER reactions. Notably, the HER activity of the Co@g-CN catalyst with adequate Co content was close to that of the Pt/Cp catalyst, effectively minimizing the overpotential. The catalytic performance of the present catalyst is also compared with previously reported works in Figure 7C and Table 3. Despite the present electrode being slightly inferior to that of Pt/Cp, it was better for the HER activity and somewhat good for the OER activity, compared to some advanced catalysts [10,18,19,22]. Moreover, most reports [19,30,31] were performed under alkaline conditions, different from the present work, which was conducted using acidic conditions. The present condition is preferable for minimizing the cost of performance by using only a cell with an acidic medium instead of two cells with separate acidic and basic media.

To determine the overall water-splitting performance, the voltage differences (ΔV) between the HER and OER were calculated by deducting HER overpotential from OER overpotential at the current density of 10 mAcm^−2^, as indicated in Figure 7D. The Co(11)@g-CN catalyst afforded a current density of 10 mAcm^−2^ at a cell voltage of 1.84 V in the acidic electrolyte, which was slightly less than that of the benchmark catalyst (Pt/Cp, 1.98 V). Although the voltage differences of some reported works at the same current density were lower [10,16,18,32,33] or similar [19] (see Table 3), the present work has merit since the water splitting was performed only in an acidic medium and used only mono-metallic conductive metal.

## 4. Conclusions

In this work, the HER and OER activities of Co@g-CN electrocatalyst were reported and compared with pristine g-CN and a benchmark catalyst (Pt/Cp). Since the encapsulation of cobalt in g-CN effectively enhanced the kinetics of water adsorption and exposed favorably active sites that can result from the increase in the surface area and the electron transfer capacity of the g-CN catalyst, the activity increased compared to that caused by the pristine g-CN. In this regard, the relevant catalyst exhibited good catalytic activities for both the HER and OER in 0.5 M H_2_SO_4_ (aq.). The optimum catalyst (Co@g-CN) displayed the better activity, which exceeded the g-CN electrode for the HER. For the OER, the same electrode achieved lower overpotential relative to the benchmark catalyst (Pt/Cp) at a current density of 10 mAcm^−2^. Furthermore, the catalyst showed better overall water-splitting cell voltage of 1.84 V to drive a current density of 10 mAcm^−2^ as a bifunctional catalyst. Additionally, the recyclability of the catalyst was better, since the cobalt in g-CN increased the active site for water adsorption. Considering the electrocatalytic activities and simple synthesis approach of the catalyst, this work provides insights to design and develop an economical and valid bifunctional catalyst for overall water splitting in acidic media. Additionally, the Co@g-CN catalyst holds great potential in the fields of electrocatalyst and energy conversion.

## Figures and Tables

**Figure 1 molecules-27-06445-f001:**
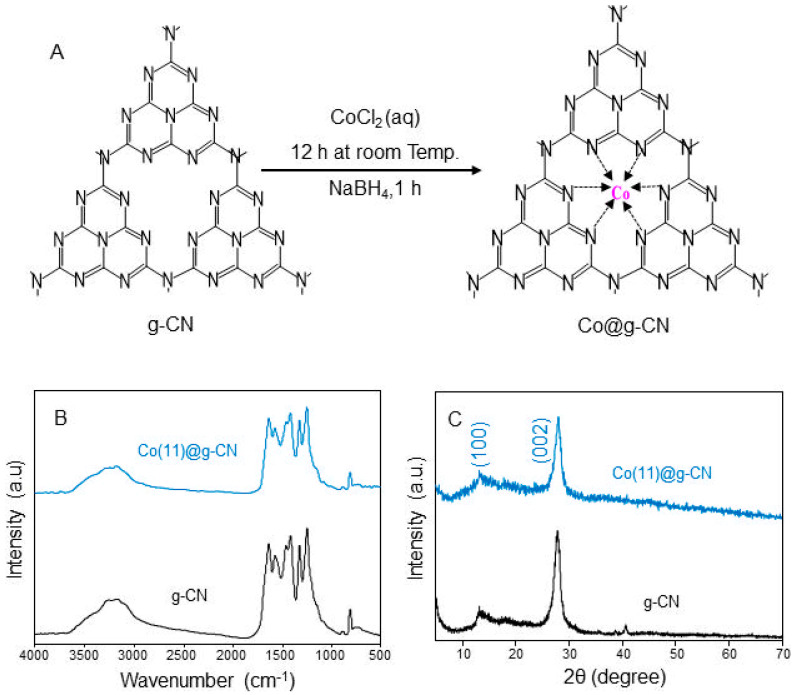
(**A**) Schematic diagram of a synthesis procedure of Co@g-CN from g-CN, (**B**) FTIR spectra and (**C**) XRD patterns of g-CN and Co(11)@g-CN.

**Figure 2 molecules-27-06445-f002:**
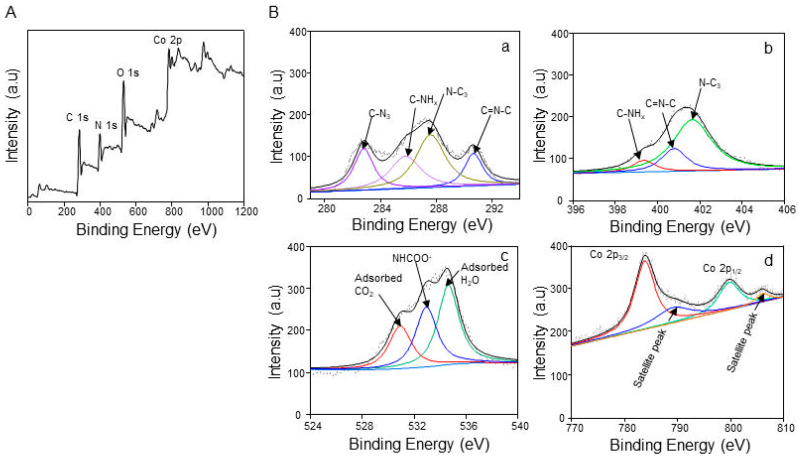
(**A**) A survey XPS and (**B**) deconvoluted fine spectra of (**a**) C 1s, (**b**) N 1s, (**c**) O 1s, and (**d**) Co 2p of Co(11)@g-CN.

**Figure 3 molecules-27-06445-f003:**
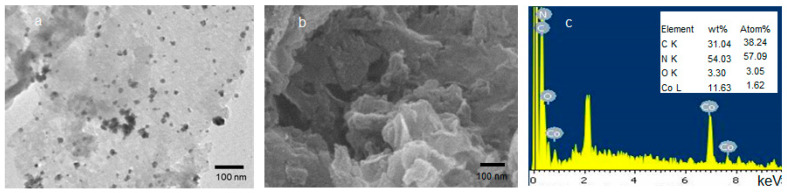
(**a**) A TEM image, (**b**) a FE-SEM image, and (**c**) EDS analysis with an inset table of the element content of Co(11)@g-CN.

**Figure 4 molecules-27-06445-f004:**
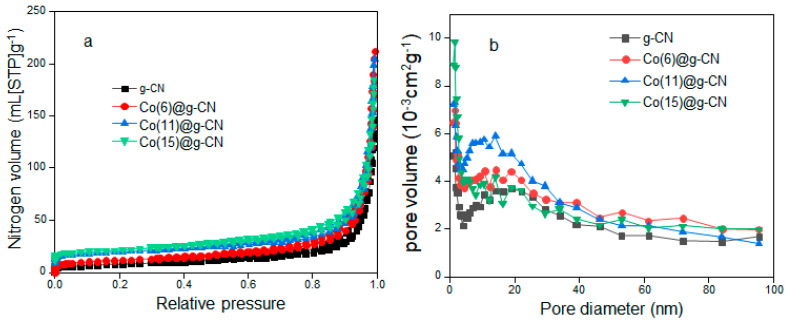
(**a**) Nitrogen adsorption-desorption isotherms of g-CN and Co(x)@g-CN (x = 6, 11, 15) and (**b**) their respective BJH plots.

**Figure 5 molecules-27-06445-f005:**
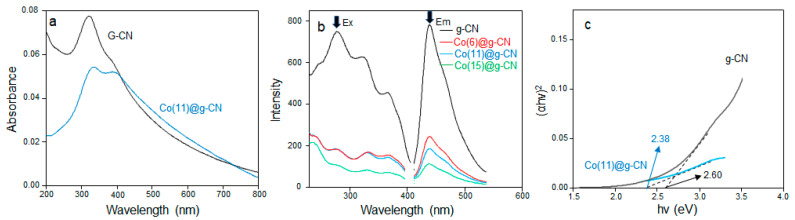
(**a**) UV-visible absorption spectra of g-CN and Co(11)@g-CN, (**b**) fluorescence spectra of g-CN and Co(x)@g-CN (x = 6, 11, 15), and (**c**) Tauc plots of g-CN and Co(11)g-CN.

**Figure 6 molecules-27-06445-f006:**
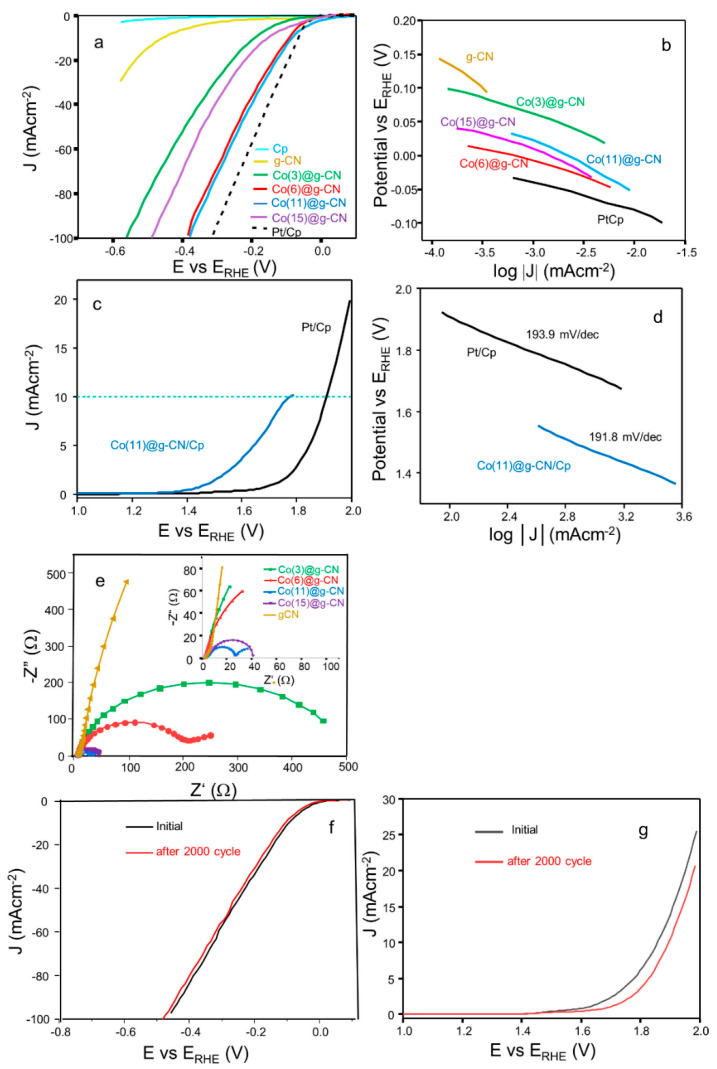
(**a**) LSV of HER activity of catalysts, and (**b**) their corresponding Tafel plots; (**c**) LSV of OER activity of Co(11)@g-CN and Pt/Cp, and (**d**) their corresponding Tafel plot; (**e**) EIS of catalysts with a magnified inset, (**f**) recyclability test of HER, and (**g**) recyclability test of OER on Co(11)@g-CN after 2000 cycles.

**Figure 7 molecules-27-06445-f007:**
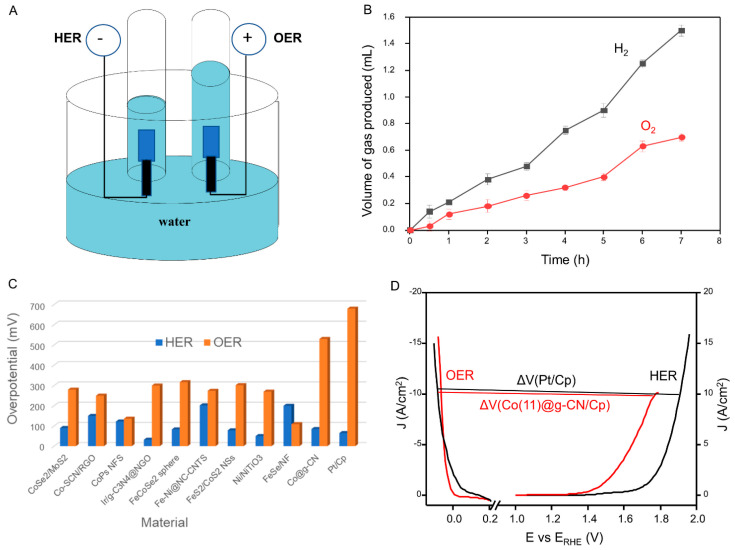
(**A**) A setup for the quantification of H_2_ and O_2_ gases by a downward displacement of water, (**B**) volumes of H_2_ and O_2_ gases evolved over time, (**C**) overpotentials of HER and OER from the current work and previous reports, and (**D**) overall water-splitting overpotential of Co(11)@g-CN (red) and Pt/Cp (black).

**Table 1 molecules-27-06445-t001:** Nitrogen adsorption-desorption isotherm analyses of different catalysts.

Catalyst	Specific Surface Area (m^2^·g^−1^)	Pore Volume (cm^2^·g^−1^)	Pore Diameter (nm)
g-CN	28.34	0.224	31.64
Co(6)@g-CN	41.26	0.293	28.44
Co(11)@g-CN	46.05	0.299	25.97
Co(15)@g-CN	38.12	0.266	27.87

**Table 2 molecules-27-06445-t002:** Electrochemical parameters of different catalysts.

Catalyst	HER Overpotential at 10 mVcm^−2^ (mV)	HER Tafel Slope (mV/dec.)	OER Overpotential at 10 mVcm^−2^ (mV)	OER Tafel Slope (mV/dec.)	Rs (Ω)	Rct (Ω)
g-CN/Cp	459	106.4	-	-	4.45	6057.8
Co(3)@g-CN/Cp	88	110.2	-	-	1.92	517.3
Co(6)@g-CN/Cp	96	69.9	-	-	2.15	203.8
Co(11)@g-CN/Cp	85	44.2	530	191.8	2.96	27.0
Co(15)@g-CN/Cp	65	102.4	-	-	2.45	40.8
Pt/Cp	65	38.1	680	193.9	-	-

**Table 3 molecules-27-06445-t003:** Parameters of electrochemical water splitting by bifunction catalysts.

Catalyst *	Medium	HER Tafel Slope (mv/dec.)	HER Overpotential at 10 mVcm^−2^ (mV)	OER Tafel Slope (mv/dec.)	OER Overpotential at 10 mVcm^−2^ (mV)	Overpotential Voltage Difference (ΔV) (V)	Reference
Co-SCN/RGO	Basic	94	150	96	250	1.63	[10]
Ir/g-C_3_N_4_/NGS	Acidic	22	28	72.8	300	1.56	[16]
^0^D-^2^D CoSe_2_/MoSe_2_	Basic	84.8	90	86.8	280	1.63	[32]
CoP NFs	Acidic, Basic	35.5	122	49.6	323	1.65	[18]
FeCoSe_2_@g-C_3_N_4_	Acidic, Basic	-	83	-	360	-	[20]
MIL-88-Fe/Ni	Basic	33.8	202	45.5	274	-	[31]
FeS_2_/CoS_2_	Basic	44	78.2	42	302	1.47	[33]
Ni/NiTiO_3_	Basic	118	50	62.2	270	1.65	[34]
FeSe@NF	Basic			145	200	1.85	[19]
Pt/Cp	Acidic	38.1	65	193.9	680	1.98	This work
Co(11)@g-CN	Acidic	44.2	85	191	530	1.84	This work

* SCN = sulfur-doped graphitic carbon nitride, RGO = reduced graphene oxide, NG = nitrogen-doped graphene, NF = nano frame, MIL-88 = metal organic framework of Fe/Ni, and NF = Ni foam.

## Data Availability

All data has been made available through the manuscript.

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
