# Peer review of "Cobalt Incorporated Graphitic Carbon Nitride as a Bifunctional Catalyst for Electrochemical Water-Splitting Reactions in Acidic Media"

_molecules, 2022, doi:10.3390/molecules27196445_

Round 1

Reviewer 1 Report

In this work, a series of Co@g-CN was synthesized and fabricated into carbon paper as bifunctional catalyst for water electrolysis. What is noteworthy is that the Co(11)@g-CN exhibits good activity and stability both in HER and OER. The composites consisting of cobalt encapsulated in g-CN effectively enhanced the kinetics of water adsorption and exposed favorably active sites which can result from the increase of the surface area and the electron transfer capacity of g-CN catalyst, which is why it has superior activity. This work is recommended for publication after some minor revisions.

1 In Introduction part, the word “hydrogen fuel cell” is not accurate. Hydrogen fuel cell is designed to consume H2/O2 and generate power.

2 The author used a platinum wire as a counter electrode in electrochemical tests which is inappropriate. Because Platinum is highly active for HER, and using a platinum wire as counter electrode will cause platinum dissolving and the dissolving platinum will deposit in working electrode and enhance its activity.

3 Please explain why the CVs tests is between 0.1 to -0.11V vs RHE.

4 Please describe what is ERHE in the formula “η[V]=0.000- ERHE

5 Figure 3, 4 and 5 is not satisfied this journal’s demand.

6 The mark number before “Electrochemical water splitting activity of Co@g-CN catalysts” is missing, please correct.

7 The benchmark catalyst for OER should be commercial IrO2, the comparison should be Co@g-CN and commercial IrO2 not Pt/Cp.

Author Response

attached a file.

Reviewer 2 Report

The present work shows promising potential as an alternative to the current commercial Pt/Cp. However, there are several issues with the current work that need to be addressed before it can be accepted for publication. 

Introduction, L5: confusing sentence. Fuel cell is not a water splitting generator. Fuel cell consumes hydrogen to produce water. 

P2, L1: under trial

what is the mode for FTIR measurement? is it transmission mode, ATR, or DRIFTS? 

Please indicate in Fig 1c the peaks for (100), (002), and so on. 

Please repeat the XRD measurement with a different X-ray source. Co has strong fluorescence effect under Cu source and therefore has very low sensitivity to Co. 

The XPS analysis doesn't match. While O1s shows adsorbed CO2 and NHCOO-, these peaks are not observed in C1s. 

The UV-Vis spectra and Tauc plot bandgap estimation should take into consideration the background absorption. g-CN bandgap should be about 2.7eV. 

how does the catalysts conductivity in the current study compare to the Pt/Cp benchmark? the data is missing from the EIS figure. 

If Co is incorporated into the CN structure, why does the TEM show Co nanoparticles of ~10-20nm in diameter? 

Author Response

Attached a file.

Round 2

Reviewer 2 Report

The authors have addressed most comments satisfactorily. However, the UV-Vis analysis for band gap measurement is incorrect. Please refer to the application note from Shimadzu below on how to determine the band gap. 

https://www.shimadzu.com/an/sites/shimadzu.com.an/files/pim/pim_document_file/applications/application_note/13524/jpa114010.pdf

Author Response

I attached reply.
